# communications
# engineering

# Nano scale instance-based learning using non-specific hybridization of DNA sequences

Yanqing Su [1,5], Wanmin Lin[1,5], Ling Chu[1], Xiangzhen Zan[1], Peng Xu[1,2,3 ✉], Fengyue Zhang[4 ✉], Bo Liu[4] & Wenbin Liu [1,2 ✉]

DNA, or deoxyribonucleic acid, is a powerful molecule that plays a fundamental role in storing and processing genetic information of all living organisms. In recent years, scientists have harnessed hybridization powers between DNA molecules to perform various computing tasks in DNA computing and DNA storage. Unlike specific hybridization, non-specific hybridization provides a natural way to measure similarity between the objects represented by different DNA sequences. We utilize such property to build an instance-based learning model which recognizes an object by its similarity with other samples. The handwriting digit images in MNIST dataset are encoded by DNA sequences using a deep learning encoder. And the reverse complement sequence of a query image is used to hybridize with the training instance sequences. Simulation results by NUPACK show that this classification model by DNA could achieve 95% accuracy on average. Wet-lab experiments also validate the predicted yield is consistent with the hybridization strength. Our work proves that it is feasible to build an effective instance-based classification model for practical application.

[1] Institution of Computational Science and Technology, Guangzhou University, Guangzhou, China. [2] Guangdong Provincial Key Laboratory of Artificial Intelligence in Medical Image Analysis and Application, Guangzhou, China. [3] School of Computer Science of Information Technology, Qiannan Normal University for Nationalities, Duyun, China. [4] Institute of Medical Artificial Intelligence, Binzhou Medical University, Yantai, China. [5] These authors contributed equally: Yanqing Su, Wanmin Lin. ✉email: gdxupeng@gzhu.edu.cn; fyzhang@bzmc.edu.cn; wbliu6910@gzhu.edu.cn

DNA has been well known to play an essential role in storing and transmitting genetic information to sustain the development, growth, and function of all living organisms. From the perspective of information processing, DNA molecules have three unique properties: high density, long durability, and low energy consumption[1]. These advantages have attracted scientific communities to explore new nanoscale paradigms to cope with the challenges confronted by electronic information technologies. In the past 30 years, DNA computing and DNA storage have been two emerging disciplines in this field[2]. DNA computing was initially proposed to solve a famous combinatorial optimization problem: TSP (Traversal Salesman Problem)[3]. However, studies in this direction were frustrated by the exponential amounts of DNA sequences produced.

Later, researchers turned towards DNA SDR (Strand Displacement Reaction) and algorithmic self-assembly, which could autonomously process information encoded in the input DNA sequences[4–7]. The programmability and biocompatibility of such complex molecular machines and systems have potential applications in biosensing[8,9], drug delivery[10], disease diagnostics[11,12], and pattern recognition[13]. In 2011, Qian et al. implemented a neural network-like computation using DNA seesaw gates which could perform weight multiplication, integration, and thresholding. They demonstrated that such a neural network-like gate could simulate a Hopfield memory to perform four 4-bit pattern classifications[13]. In 2017, Cherry et al. further designed a more extensive neural network to recognize $10 \times 10$ patterns for handwritten digits '1' to '9' in the MNIST dataset[14]. The classification process was actually to match the sequences of two images pixel by pixel. Xiong et al. implemented a CNN (Convolutional Neural Networks) via a DNA switching gate architecture, where the CNN was first trained in silico, and then the obtained weights were encoded in the hairpin stem of each DNA switching gate. They demonstrated that it could classify up to thirty-two $12 \times 12$ patterns[15]. Lopez et al. implemented a linear SVM classifier based on a novel class of DNA probes that directly take the disease-related RNA transcripts as inputs. They validated such SVMs in two applications: one for early cancer diagnostics and another for differentiating viral and bacterial respiratory infections[16]. Yin et al. implemented a DNA-framework-based classifier based on programmable atom-like nanoparticles to perform prostate cancer taxonomy using multidimensional datatypes including mRNA, miRNA, protein and small molecule[17].

As remarked in a survey by Reif et al.[18], the SDR-based classifier is still in its infancy stage and it needs a long way to develop general-purpose architectures instead of just working for the case under consideration. Currently, the size of the implemented neural network-like computation has been noticeably low, owing to the scale and complexity of implementing such systems. Their computing capability still suffers from many problems, such as undesired leak reactions, considerable reaction delay, and reaction cross-talk. Last but not least, the designed computing gate is just a one-shot device used up in one forward pass of the input.

In the above computing processes, the specific hybridization between DNA molecules plays a fundamental role where non-specific hybridizations may lead to unexpected results in the multistage cascading reactions[18,19]. Recently, Bee et al. harnessed the non-specific hybridization to execute a similarity search over a DNA database of 1.6 million images. They trained a VGG16 deep learning network as an image-to-sequence encoding so that queries preferentially bind to visually similar targets[20]. In their work, the non-specific hybridization actually achieved a kind of parallel similarity computing in large-scale DNA molecules[21]. This motivates us to adapt it for solving a non-parametric learning task known as instance-based learning[22], which predicts a new data instance based on its similarity to previously observed instances. The key advantages of this approach lie in its simplicity, robustness, and its potential to achieve optimal performance across a wide range of challenging classification problems. In practice, its implementation is usually challenged by the need for enormous memory space and computing costs in similarity comparison. However, these two challenges may disappear at the molecular scale.

In this paper, we demonstrate the construction of an instance-based classifier by DNA molecules to recognize the handwritten digits in the MNIST database[23]. A LeNet-5 CNN network[23] is used to extract image features, while an encoder and a predictor are used to map similar images into similar DNA sequences. According to NUPACK[24], an authoritative software suite in the field of DNA computing and storage[21,25], the proposed classifier has the potential to achieve an average accuracy of 95% in dry-lab. A wet-lab experiment with 50 instances validates that the predicted yield by NUPACK is consistent with the hybridization strength between the query sequence and instances. In sum, both the simulation and wet-lab results demonstrate that it is possible to build a practical instance-based learning model by DNA molecules which could accommodate millions of instances and perform parallel computing.

## Results

**The workflow of the image encoding architecture.** Figure 1a outlines the three main blocks of our image encoding architecture, a LeNet-5 backbone for feature extraction, an encoder to map feature vectors to DNA sequences, and a predictor to optimize the encoder effectively. LeNet-5 is a classic CNN consisting of seven layers, specifically designed for handwritten digit recognition. Given a single-channel $28 \times 28$ grayscale image (in MNIST), the output of the second fully connected layer (FC2) is used as its feature vector.

To build a DNA instance-based classifier for the MNIST dataset, the critical step is to encode digit ('0' to '9') images so that similar images should be encoded by similar DNA sequences. That is, the reverse complement sequence of one image should be highly hybridized with the sequences of the same class images (the same digit image, judged according to the MNIST label), and less likely with those of the other class images. Then, the query label is assigned to the class with the highest hybridization strength.

To optimize the encoder for this goal, we need an indicator to quantitatively reflect the hybridization degree of pairwise DNA sequence along with a suitable tool to predict it effectively. The "yield" is defined as the ratio of the double-strand DNA (dsDNA) concentration to the initial single-strand DNA (ssDNA) concentration at equilibrium[21,25]. Obviously, the closer the yield is to 1, the stronger the hybridization degree of the pairwise DNA. Yield can be accurately estimated by NUPACK. However, operations within NUPACK are not continuous and differentiable, rendering it inappropriate for the proposed CNN-like structure. Other simple approaches, such as the edit distance of sequences, cannot be treated as an accurate approximation of yield (as explained in Supplementary Note 1). Therefore, a two-layer CNN is employed as the predictor to approximately substitute NUPACK in our architecture.

The encoder, composed of two linear layers with softmax activation, maps a 50-dimensional (50-D) feature vector into a $4 \times 59$ tensor. Each row of the tensor corresponds to a nucleic acid base: A (Adenine), T (thymine), C (Cytosine) and G (Guanine). The row with the maximum value in each column is determined as the output base. Based on this rule, the $4 \times 59$

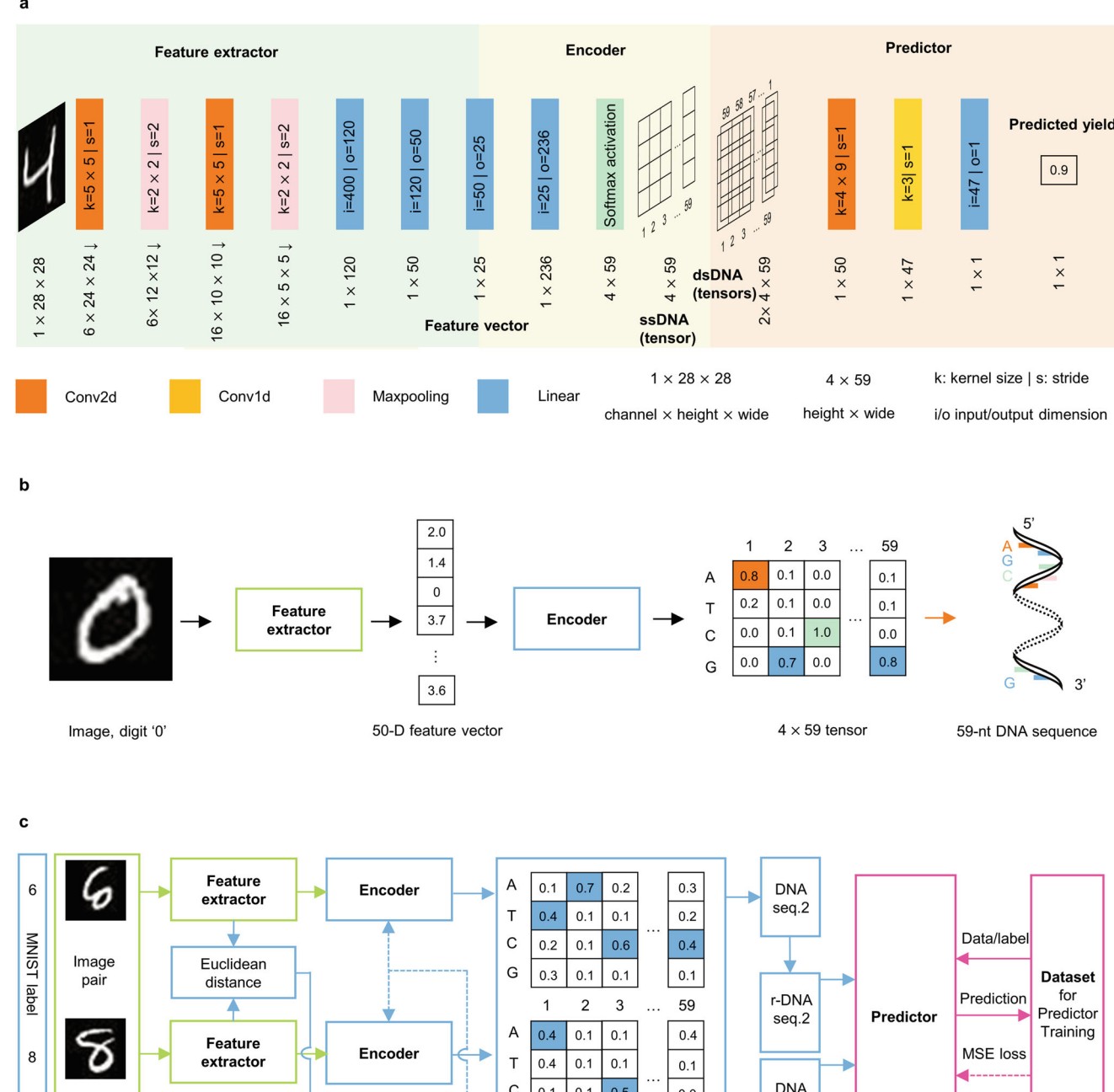

**Fig. 1 Image encoding architecture. a** Detail structure of the feature extractor, encoder, and hybridization predictor. **b** An image encoding example. Colored cells in the encoder output mark the maximum value in this column. **c** The image encoding architecture. There are two identical feature extractors and encoders shown in the figure to synchronize the feature extractor, the encoder and the predictor. Three training stages are shown in three colors.

tensor is converted to a 59-nt (59-nucleotide) DNA sequence (Fig. 1b).

Figure 1c shows the workflow of the encoding architecture, and the training process includes three stages: First, train the LeNet-5 backbone to classify the 60,000 images in the MNIST dataset.

Second, train the predictor with 300,000 DNA sequence pairs whose yields are labeled by NUPACK. Finally, train the encoder based on the similarity of the feature vectors, their MNIST labels and the predicted yield. Details of training and encoding can be found in the Method section.

**The encoding performance**. The MNIST dataset is composed of two mutually exclusive subsets: the training set of 60,000 images and the testing set of 10,000 images. Figure 2a shows the clustering result of the training images on a 2-D plane by t-SNE[26], where the left is based on the LeNet-5 feature vectors and the right is on the encoded DNA sequences. In both cases, images within the same class label are almost clustered in the same group. However, their neighboring relationship in the feature vector space may be different from that in the DNA sequence space.

This is because the training process considers both the MNIST labels and the similarity of feature vectors. This difference in the neighboring relationship can be observed for the four-pair images in Group A and B. In general, the encoded DNA sequences could reflect the similarity of images to a high degree.

Figure 2b shows the distribution of yield according to the Euclidean distance of their feature vectors. Most of the yield is larger than 0.8 when the Euclidean distance is less than 9, and the yield is close to 0 when the Euclidean distance is larger than 17. When the Euclidean distance ranges from 9 to 16, about one-third is close to one and the rest is close to zero. Figure 2c presents four pairs of images within each distance interval to illustrate the relationship between visual similarity and their Euclidean distance. When the Euclidean distance is larger than 9, most of the images tend to be different. All in all, the yield distribution reveals that the encoder assures similar images usually have a large hybridization yield (>0.7) while those dissimilar have a low yield (<0.2).

Figure 2d shows the top 50 nearest neighbors of a query sequence of digit '5' according to the predicted yield by NUPACK. As expected, images of digit '5' are the first majority which is 46%. The second and third majority comes from digit '9' and '6' which is about 14% and 12%. Further, most of the images of digit '5' are in the top 30. This means that the total hybridization yield of the query sequence with those of digit '5' should be significantly larger than that with other digit images. This demonstrates that the total yield of the query sequence with each digit label could serve as an indicator of the similarity degree.

**Simulated classification performance by NUPACK**. We take all images in the MNIST training set as instances and randomly draw 1000 images (100 for each digit) from the MNIST testing set as queries to verify the performance of the proposed instance-based classifier. Simulation experiment (dry-lab experiment) protocol can be found in "Method" section.

Figure 3a, b show the yield distribution of these query sequences with those in the same class (a) and those from other classes (b). (In the remainder of the manuscript, the same/different class means images in this class have the same/different MNIST label. Class 0, 1, ..., 9 means images' MNIST labels of the class are 0, 1, ..., 9 and digit '0', '1', '2', ..., '9') Comparing Fig. 3a, b, we could see that these query sequences usually have larger yield with their own class sequences (>0.8) and smaller yield with other class sequences (<0.1). These observations indicate that the query sequences tend to have higher yields with their own class instances than with others.

Figure 3c presents the average accuracy for each digit class and the overall accuracy is about 95%. Only class 4 has the least accuracy (86%), and this is consistent with the yield distribution in Fig. 3a, b. That is, the query sequences have a quite low yield with their own class sequences and a relatively larger yield with other class sequences. The main reason may be attributed to its quite low encoding quality. Back to the clustering results in Fig. 2b, we can see that the instances of class 4 are scattered in

two sub-clusters surrounded by six other clusters. Furthermore, Fig. 3d presents the 51 misclassified query images, most of which are irregular handwritings. Their detailed predicted yields for each class can be found in Supplementary Note 5.

**Experimental validation of the hybridization yield**. To verify the consistency between the predicted yields by NUPACK and real hybridization outcomes, we designed a small classifier with 50 instances for each class. We select ten queries (one for each class, Fig. 4a) which have relatively low yields with their own class instances. Figure 4a shows the simulated yields by NUPACK between these queries with the 50 instances for each class. The predicted yield for each query with its own label is always the largest (see the diagonal values in each row). That means the classifier could correctly recognize the digit written even though they may have some degree of similarity with other classes.

Furthermore, the query sequences for 5, 7, and 8 have relatively high yields, with the 50 instances for class 3. We conduct a wet-lab experiment to measure the real hybridization strength between 10 queries and 50 instances of class 3. Figure 4b shows the measured fluorescence intensity, and the larger the intensity, the higher the degree of the hybridization would be (See Method section). For each case, we repeat the three-time experiments. Columns A and B are used as control groups for TE buffer and TE buffer added with fluorescent dye, respectively. The rest of columns correspond to query sequences for digit '0' to '9' (Q0 to Q9). We could see that column F (Q3) has the largest fluorescence intensity than others. That is, Q3 has the highest hybridization strength with the instances of class 3. And columns K(Q8), J(Q7), and H(Q5) have a relatively larger intensity. These observed fluorescence intensities are consistent with the average predicted yield for the 10 query sequences in Fig. 4a. This demonstrates that the predicted yield could reflect the hybridization strength, and it is feasible to build a large classifier with DNA sequences which is composed of tens of thousands of instances.

**Discussion**
In this paper, we propose an instance-based classifier by non-specific hybridization of DNA sequences. Images in the MNIST dataset are encoded into DNA sequences by a deep learning network in Fig. 1. Then the DNA sequences representing instances are synthesized and those belonging to the same class are grouped in one test tube. The reverse complement sequences of the query image are used to hybridize with those instances simultaneously. Finally, the one with the highest hybridization strength is assigned as the output. Simulation results by NUPACK and the wet-lab experiment demonstrate the feasibility of the proposed method. Compared with the SDR-based classifier[14], the proposed classifier by non-specific hybridization has the following advantages.

**Firstly, it is more robust**. The implementation of the cascade of the seesaw DNA gate is not only time-consuming but also sensitive to biochemical reaction conditions, such as the concentration gradient of substrates, temperature, and perfect hybridization. On the contrary, the non-specific hybridization process in the proposed classifier requires no these rigorous requirements and only needs one round of hybridization.

**Secondly, it is easily implemented and more efficient**. The cascade of SDR reactions is very complicated and laborious, which has to screen the output DNA strands as the input of the next round of reaction. The proposed classifier could be implemented on a solid surface[27] on which millions of samples can be organized according to their class labels. The fluorescence

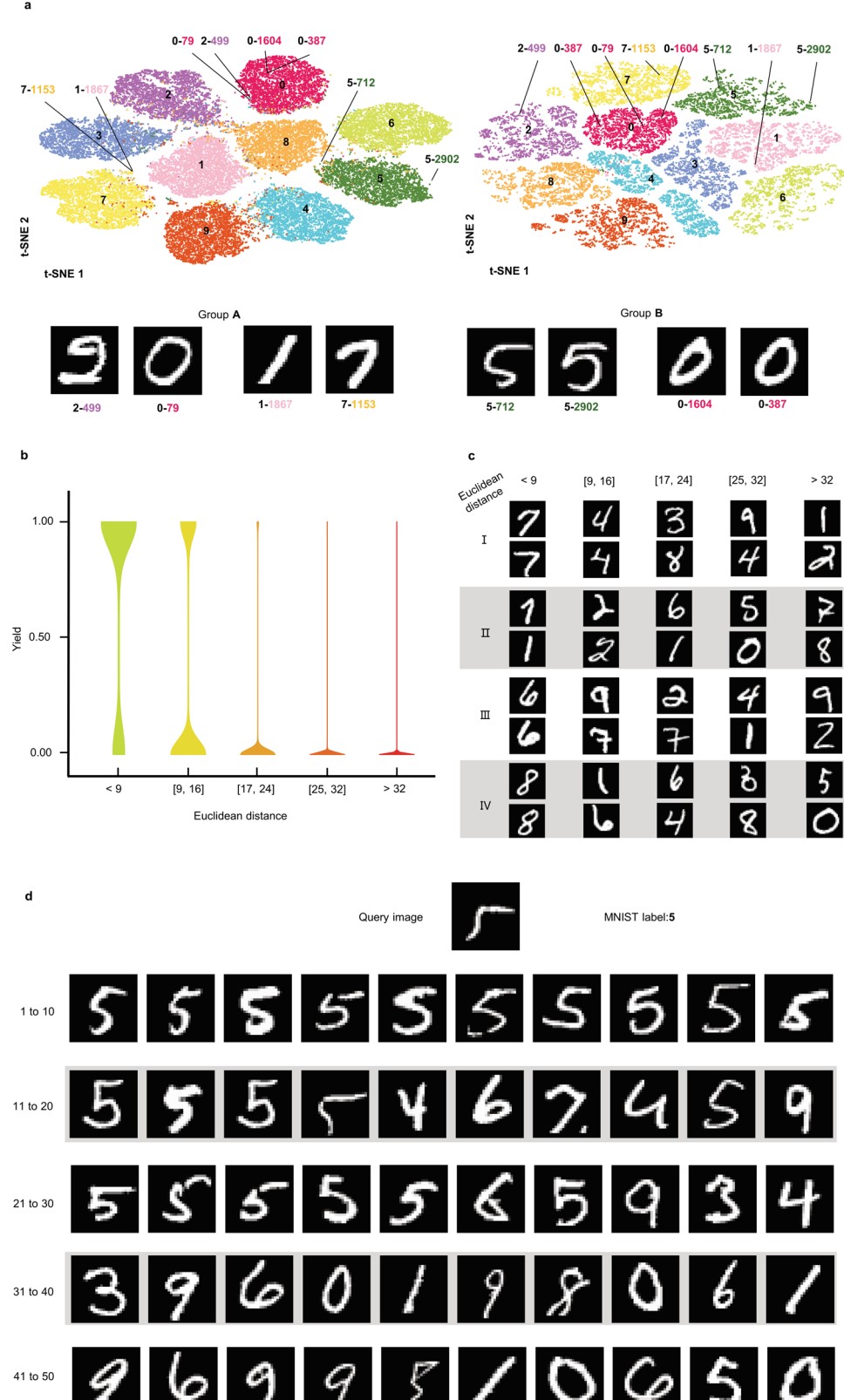

**Fig. 2 The encoding performance. a** 2D t-SNE projection of image feature vectors and their encoded DNA sequences. Images or sequences from the same class are represented by dots in the same color. **b** Encoding performance. A violin graph of the yield in different Euclidean distance intervals. **c** Example images in each Euclidean distance interval in **b**. **d** The top 50 nearest neighbors of a query sequence of image '5'.

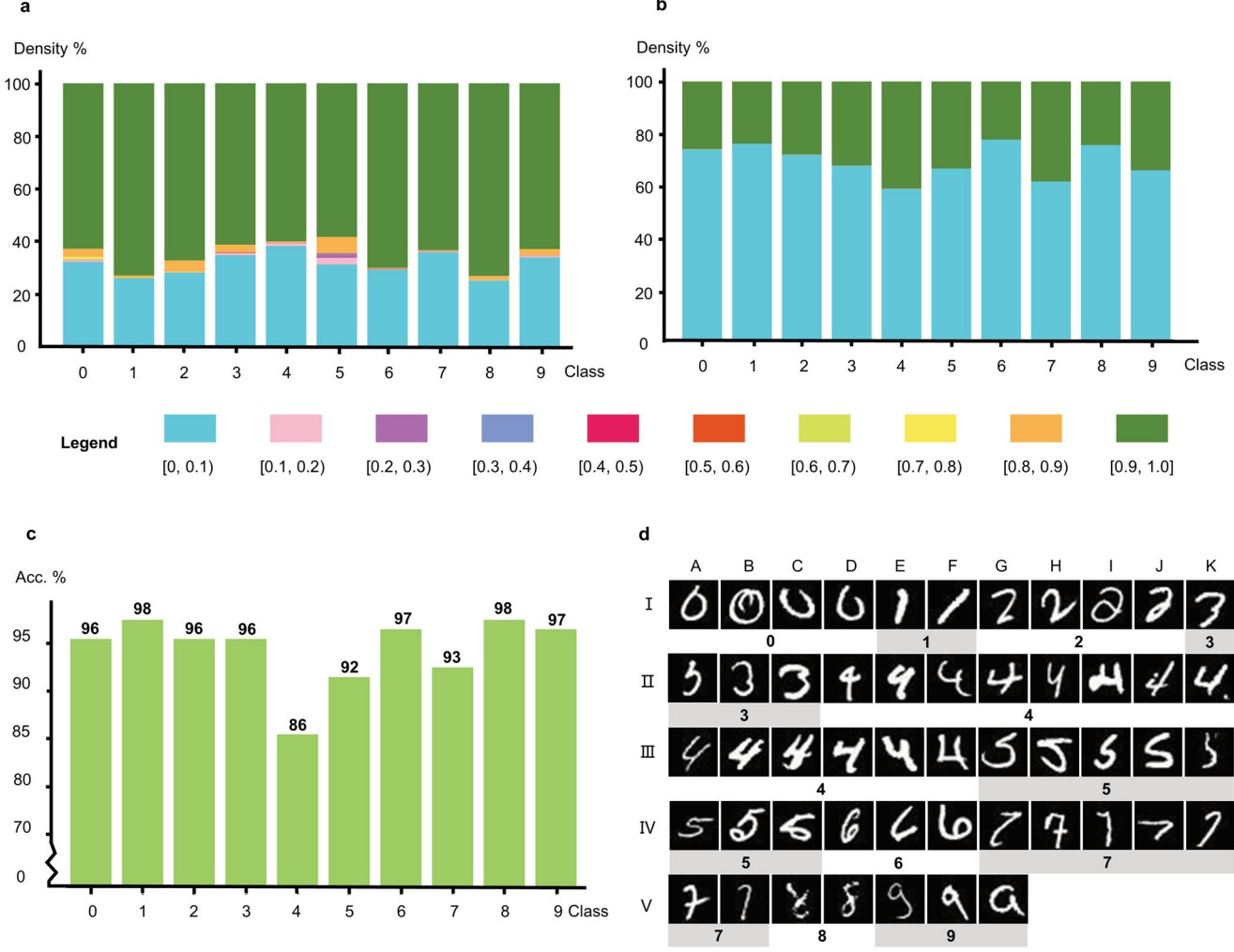

**Fig. 3 Simulated classification performance of 1000 query sequences by NUPACK. a** The average yield distribution between the query sequences and their own class sequences. **b** The average yield distribution between the query sequences and other class sequences. **c** The average accuracy for each digit number. **d** The misclassified images and their corresponding labels.

intensity of hybridization can be directly read out by an optical scanning machine. Furthermore, the non-specific hybridization relieves the computation burden of calculating the similarity between the query sample and the training samples. Combined with the nanoscale of DNA molecules, it provides an easy way to construct a classifier when facing massive training samples.

**Thirdly, it can be flexibly extended to other larger complicated datasets**. To extend the SDR-based classifier for more classes, the complexity of the SDR computation circuit had to be increased dramatically and more likely to lead to reaction failure[13–17]. However, the proposed classifier could accommodate problems with a hundred or more classes. For example, considering the applications for images in OpenImages[28], we could adopt the encoding network architecture in Bee et al.'s work to encode images with more channels and class labels[21]. Further, the excellent performance of Transformer[29] also provides a flexible embedding techniques to encode different objects such as the image[30] and the text[31].

Finally, we should point out that the encoding quality of images plays an essential role in the reliable performance of the proposed classifier. For future large-scale applications with hundreds/thousands of classes, developing a highly efficient

encoding architecture is urgently needed. In addition, the influence of the secondary structures of ssDNA is another critical issue to be considered, which may affect the non-specific hybridization and result in off-target effects. In the future, we will focus on these two problems to consolidate the application of the proposed classifier for more complicated scenarios.

## Method

**Feature extraction**. To extract image features by LeNet-5 backbone, we first modify the dimension of FC2 to 50. Subsequently, the fine-tuned LeNet-5 is used to perform MNIST classification tasks to obtain optimal parameters for all layers. Then, the outputs of FC2 are used as the image feature vectors (50-D). All parameters of the network keep unchanged during feature extraction process.

**Predictor training**. A dataset of 300,000 pairs of DNA sequences with a length of 59-nt is randomly generated for predictor training. To ensure a more realistic representation, homopolymers are excluded from the dataset, which refers to any instance where three or more consecutive identical bases (e.g., AAA, TTTT) occur in the sequence. Each sequence pair is then labeled with a simulated yield from NUPACK. Then fine-tune the

**a**

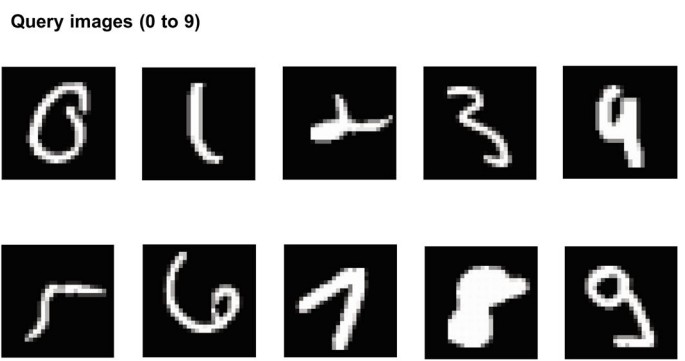

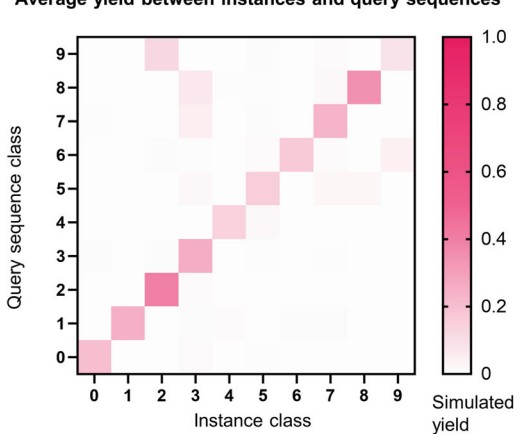

**b**

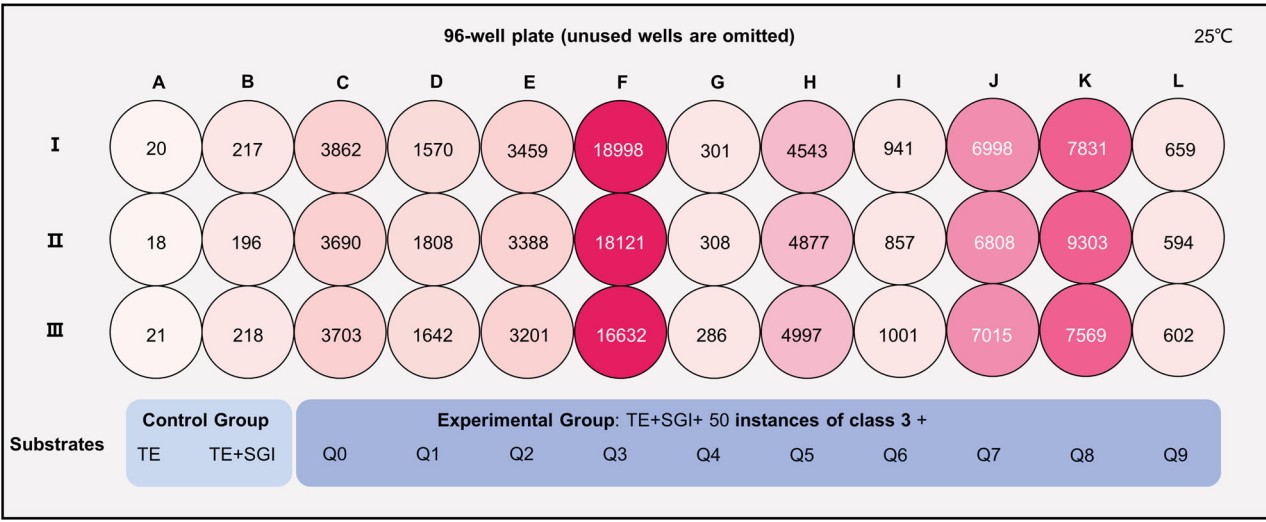

**Fig. 4 Experimental validation of the hybridization yield. a** Dry-lab experiment result when the proposed classifier faces a hard-to-classify situation. Yields at the diagonal are always significantly larger than those at the same row for each query sequence, which indicates the classifier can smoothly complete the classification tasks of fuzzy queries. **b** The fluorescence intensity (repeated three times). Columns A and B are the control experiments with TE buffer and TE buffer added with fluorescent dye. Columns C to L are for queries of digit '0' to '9' hybridization situation with the 50 instances of digit '3'.

dataset to remove repeat pairs and ensure an even distribution of pairs with different yields. Sequence pairs are adapted into One-hot form and fed into the predictor. The estimated yield, along with the pairs' labels are used to compute the mean square error loss (MSELoss) for each batch. The parameters of the predictor are adjusted via gradient descent to minimize MSELoss.

**Encoder training**. To train the encoder for the two goals, three issues as follows should be exhaustive discussed.

How to measure image similarity? The similarity of image pairs is primarily measured by Euclidean distance of their feature vectors and secondary consideration, their MNIST labels. Ideally, similarity should be judged solely by MNIST labels, which encourages the encoder to map images in the same class into high-yield sequences. However, there are some neighboring exceptions with very different labels (such as images in Fig. 2a Group A). Considering only one of the two leads to excessive regularization for the encoder, ultimately causing a training failure.

Sync problem with the feature extractor, the encoder and the predictor. As previously discussed, the predictor estimates yield of pairwise DNA sequence while the feature extractor and the encoder process one image at a time. To address the asynchronous problem, we call the feature extractor and the encoder twice to obtain the first and the second ssDNA to catch up the pace with the predictor. (That is why there are two same feature extractors and encoders in Fig. 1b but in our code, there is an encoder and a feature extractor in the proposed architecture.) Sequence pairs passed to the predictor are comprised of the first sequence and the reverse complement of the second sequence. The output of the encoder, estimated yield, Euclidean distance of feature vectors and MNIST labels are then processed by loss function of the encoder.

Loss function of the encoder. The encoder loss is composed of two parts: encoding loss and sequence loss.

$$Encoder\_Loss = Encoding\_Loss + Sequence\_Loss \quad (1)$$

To penalize the ambiguous output of the encoder, the encoding loss is defined as:

$$Encoding\_Loss = \sum_{i=1}^{59} Encoding\_Loss_i \qquad (2)$$

$$Encoding\_Loss_i = \begin{cases} Cross\_Entropy(Y_i, T_i), & \text{if } max(Y_i) < 0.5 \\ 0, & \text{otherwise} \end{cases} \qquad (3)$$

Where $Y_i$ indicates each column of the encoder output. $T_i$ is generated from $Y_i$. Calculating cross-entropy when $Y_i$ with maximum value < 0.5 is to avoid an over-solid punishment for the encoder. Details are available in Supplementary Note 2.

Sequence loss supervises the encoder to translate similar/dissimilar images to high/low yield sequences. The sequence loss is then defined as:

$$Sequence\_Loss = \begin{cases} 1, & if\,(E < T_1 \text{ and } Y < T_2 \text{ and } L_1 = L_2)\,or \\ & (E \geq T_1 \text{ and } Y \geq T_2 \text{ and } L_1 \,! = L_2) \\ 0, & if\,(E < T_1 \text{ and } Y \geq T_2)\,or \\ & (E \geq T_1 \text{ and } Y < T_2)\,or \\ & (E \geq T_1 \text{ and } Y \geq T_2 \text{ and } L_1 = L_2) \end{cases} \qquad (4)$$

In which, $E$ is the Euclidean distance of the corresponding feature vector, $L_1$ and $L_2$ are the MNIST label of the encoding images, $Y$ indicates the estimated yield reported by the predictor. $T_1$ and $T_2$ are different thresholds, in this work, $T_1 = 16$, $T_2 = 0.8$. A flowchart of the sequence loss is available in Supplementary Fig. 2b.

Parameters of the feature extractor and the predictor remain unchanged during the whole encoder training phase.

**Sequence encoding**. Unlike training phase, images from MNIST testing set are processed by the feature extractor and the encoder "one by one". The predictor is suspended since there is no need to calculate loss.

**DNA instance-based classifier**. During the training phase, all labeled samples (in MNIST training set) are synthesized as instances and stored in tubes by their labels. In the prediction phase, query samples are encoded to DNA sequences by the proposed architecture other than the predictor. Each reverse complement of the query is used as the probe to add to tubes in turn. By analyzing the yield between the probe and instances in each tube, the query is assigned to the class that exhibits the highest yield.

In dry-lab, NUPACK can report yield between certain pairs of DNA sequences. Thus, we record the sum of yield between the query and instances in each tube to indicate the hybridization degree. The label of the query is assigned to the class with the highest yield sum. But in wet-lab, it is trivial to observe the hybridization of a specific dsDNA. At the same time, the fluorescence intensity can accurately reflect the hybridization degree between the query and all instances in the tube. Therefore, the label of the query is assigned to the class with the highest fluorescence intensity.

**NUPACK protocol**. A brief introduction of the parameters can be found in Supplementary Note 3. For both predictor training and dry-lab experiments, parameters of NUPACK are set as followed:

1. Temperature: 25 °C.
2. Initial concentration: 1 nM;
3. Max complex size: 2;
4. Model options: default.

**Materials and equipment**. All DNA sequences (10 queries and 50 instances of digit '3') for the wet-lab experiment are purchased from SBS Genetech Co.Ltd. and purified by Ultra-Polyacrylamide Gel Electrophoresis. They are dissolved in TE buffer (Beijing Solarbio Science & Technology Co., Ltd.) and stored at −20 °C. The SYBR GreenI (SGI), purchased from MCE (MedChemExpress, USA), is stored at −20 °C. Cytomics FC 500 (Beckman Coulter, USA) is the instrument to measure fluorescence intensity.

SGI is an asymmetrical cyanine dye used as a nucleic acid stain to quantify dsDNA[32,33]. The stain preferentially binds to dsDNA, which DNA-dye-complex best absorbs 497 nm blue light ($\lambda$max = 497 nm) and emits green light ($\lambda$max = 520 nm). In this paper, the fluorescence intensity is proportional to the percentage of hybridized dsDNA, which is, in turn, determined by the similarity between two sequences and whether they belong to the same class.

Cytomics FC 500 is used to measure and record the fluorescence intensity. The parameters are: 25 °C, medium speed vibrating plate for 20 s, incubation for 10 min, emission length of 485 nm, and absorption wavelength of 520 nm.

**Validation experiment of hybridization yield**. DNA sequences are dissolved in TE buffer solution (50 mM pH 8.0) and diluted to 10 μM. We add 10 μL instance (10 μM) to each of the 10 centrifuge tubes (indicating digit '0' to digit '9'). Subsequently, each query sequence (in Supplementary Note 4) is added into the corresponding tube. Vortex mixed, and heated at 95 °C for 10 min to open the secondary knot of ssDNA. Slowly cool to room temperature and dilute with TE buffer solution. The final concentration of instances and the query is 50 nM, and store them at 4 °C for standby.

Then, add 100 μL dsDNA (50 nM) to a cell of microplate, followed by 10 μL SGI of which the final concentration is 10× SGI and excessive. TE buffer only and TE buffer with SGI are taken as control blanks. There are three repeats for each reaction.

## Data availability

MNIST images are publicly available via the National Institute of Standards and Technology and also available with our models of the feature extractor, the encoder and the predictor at: https://github.com/yanqingsugzhu/nano-instance-based-classifier.

## Code availability

The code of this study is available here: https://github.com/yanqingsugzhu/nano-instance-based-classfier.

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

## Acknowledgements

This work was supported in part by the National Natural Science Foundation of China (no. 62072128, 62272060, and 62272061), the Natural Science Foundation of Guangdong Province of China (no. 2023A1515011401), and the Open Project of Guangdong Provincial Key Laboratory of Artificial Intelligence in Medical Image Analysis and Application (no. 2022B1212010011). Our heartfelt thanks go out to Zhihong Chen and Jiahao Shen for their unique companionship and invaluable discussions during this project.

## Author contributions

Y.S., W. Liu, and X.Z. conceived the concept. W. Lin developed the prototype of the workflow and obtained image feature vectors. Y.S. developed other Python codes and performed simulation experiments. F.Z. and W. Liu. designed the wet-lab experiment, and F.Z. and B.L. performed it. Y.S. organized the results, and prepared figures and tables. Y.S. and L.C. drafted the manuscript. W. Liu. and P.X. revised the manuscript and supervised the study. All authors read and approved the final manuscript.

## Competing interests

The authors declare no competing interests.

## Inclusion and ethics

Our research does not encompass inclusion or ethics considerations, as it is solely centered on developing a classifier based on synthetic DNA molecules.
