## [Peer Review File · Communications Engineering]

Reviewers' comments:

Reviewer #2 (Remarks to the Author):

From the perspective of information technology, DNA molecules have been used to solve different information challenges and has a wide application in DNA computing, DNA self-assembly and DNA storage. This is possible because of the unique properties of the DNA namely, high density, energy efficiency and long durability. Using an instance based learning model, a classifier was constructed to recognize the handwritten digits in the MINST dataset. The result from the study show the classifier mapped similar images to similar DNA sequences. A dry and wet lab simulation was used to validate the results. The main strength of the work in showcasing the potential of building a instance-based learning model using DNA to allows for parallel computing. However, the major weaknesses of the study lies in the lack of clear motivation which poses a disconnection in the understanding of the main purpose of the study.

This work is very needful to exploit all the potential of DNA molecules in the aspect of computing. However, from a reviewer's perspective, the manuscript requires major revision for clarification and understanding. Based on my review, I find that the manuscript requires additional work before it is suitable for publication.

Comment 1: The abstract is missing “the why” or “motivation” of the study. Why is the authors proposing an instance-based learning model? It is important that this reason is explicitly stated in the abstract.

Comment 2: The introduction section of your manuscript needs a stronger, more cohesive structure to help guide readers in understanding the study's focus and purpose. As it currently stands, the connection between various topics seems disjointed and not clearly focused on the study's primary objective. Please consider the following points:

(i): It is vital to specify which among the three potential DNA applications (DNA computing, DNA self-assembly, and DNA storage) your study primarily revolves around. The significance and workings of DNA hybridization in the context of information processing problems also need to be clearly elaborated. Notably, an explanation of why non-specific hybridization is chosen for this study would provide critical insight into your methodological choices.

(ii): It would also be beneficial to provide a more comprehensive background on the existing research concerning DNA-based classification of information. I recommend that you outline notable contributions in this area and address any existing questions, problems, or hypotheses. This will not only place your work within the broader scientific discourse but also help readers understand how your study advances knowledge in this field.

(iii): Your decision to propose an instance-based learning model requires more clarification. Why is this model the most suitable for your study, and how does it address existing challenges in the field?

(iv): Throughout the introduction, ensure to weave in how the cited research informs your study and how your proposed methods aim to tackle the known challenges in the field.

The reviewer suggest a comprehensive rewriting of the introduction section to incorporate these comments, providing more clarity and coherence for readers. It will also provide a clear roadmap to the issues you aim to address and how your study intends to do so.

Comment 3: Results:

An aspect that is not clear from the text is how exactly the yield is calculated and what exactly the term “hybridization constraint” referred to is.

(i) The authors referred to the yield as a measure of how well two DNA sequences bind together, but the specifics of how this is determined from the encoded DNA sequences is not detailed in the manuscripts. The authors mentioned using NUPACK but do not specify how. The yield is an important aspect of the method and could benefit from a more thorough explanation.

(ii) Additionally, the term “hybridization constraint” seems to suggest that the training process includes some kind of control or limit related to the degree of hybridization, then again, the specifics are missing. The reviewer suggests that these areas worth further explanation to fully understand the results and its implications.

(iii) Also the mapping from feature vectors to one-hot encoded DNA sequences is intriguing but the specifics are not clear too. How are these vectors translated into DNA sequences? What does a one-hot encoded DNA sequence look like?

Comment 4: The reviewer recommends that the authors explicitly acknowledge the limitations of the proposed approach and provide suggestions for future research directions. For example: while the study used the MNIST dataset as a proof of concept, the authors failed to mention how the method could be applied to other more complex datasets. More discussion around potential use cases and how this method could be generalized would be valuable.

Final comment: The authors should thoroughly read the manuscript to correct structural and typographical errors. This will improve the readability of the manuscript

Reviewer #3 (Remarks to the Author):

The paper demonstrates an image classification method using different percentages of partial hybridization between DNA strands. The specific sequences are encoded from a feature layer extracted from a convolutional neural network (LeNet-5) trained on the MNIST database. Similar classes of images will have similar sequences and thus a greater percentage of duplex hybridization that can then be observed using fluorescent stains with dsDNA preference.

The results of the paper are interesting and novel for publication, but the manuscript requires major revisions to reach publication standards. The results are positive, and the contrast with popular strand-displacement classifiers is a welcome perspective. However, the manuscript does a poor job of communicating the methods and metrics that were used, and many sections do not feel cohesive. The manuscript overall lacks clear and direct statements. It was difficult for me to understand the objectives and have a high-level understanding of the reported work as I was reading the manuscript. There is also additional data/methods that needs to be reported or edited such that it is more clearly communicated.

In detail:

1. Some terminology and word use is frequently incorrect. I suggest the authors carefully proofread the manuscript and involve a writing advisor if necessary.

All instances of "reversal complementary" should be replaced with "reverse complement".

Line 100, "hybrid" should be "hybridize".

All instances of "et.al" should be "et al." and italicized.

Line 28, "burned the enthusiasm" is not a standard expression. There is the expression "burning enthusiasm", but I do not believe it can be changed like that. "inspired enthusiasm" would be my suggested edit.

Line 45, after introducing an abbreviation (SDR, in line 42), refrain from going back to using the unabbreviated term.

Line 53, "or synthetic biology" seems redundant to mention, as there is no other synthetic biology context ever referred to again elsewhere in the paper.

Line 74, "made prediction" should be "makes predictions"

Line 140, "at equilibrium to that at the initial concentration" reads awkwardly. "at equilibrium to their initial concentration" is my suggested edit.

Line 148&287, "tenders" should be "tend"

Line 208, "The rest columns" should be "The rest of the columns"

Line 233, "propose", in conclusion, should be "proposed"

Line 247, "requires no these" is incorrect, and I'm not sure what was intended here. Was it "requires none of these"?

Line 277, "drop" and "froze" are different tenses. At least, sentences should be in the same tense.

2. NUPACK has constraints (sequence, similarity, etc.; hard and soft) for sequence design but line 300, "constrain complex is consists of up to 2 DNA sequences" is a poorly written sentence and does not give me sufficient information to understand how those NUPACK parameters were configured.

3. In the Materials and equipments section, I would prefer if the authors used SI symbols 'nM' instead of 'nmol/ L' and 'µL' instead of 'µ L'. The units should not be spaced.

4. Aptamer is typically a term reserved for DNA that has a specific binding target. Line 315 is also the very first time that 'aptamer' is used in the entire paper, so I'm not actually sure that I know what it is trying to refer to. Same issue with 'template'. Used only twice, once on line 314 and previously only on line 228. If I had to guess, these are the training samples. Terminology should be clearly introduced and then consistently used to avoid confusing the reader.

5. "10 uL template" is also confusing to read. I believe "10 uL of the template" was meant instead.

6. It was difficult to understand how the fluorescence measurements were made and why they support the results. It is never stated that SGI preferentially binds dsDNA, which determines the fluorescence readout intensity. So the fluorescence intensity is proportional to the percentage of hybridized dsDNA, which is, in turn, determined by the similarity between two sequences and whether they belong to the same class. This all needs to be directly stated, so the reader knows just exactly how the experiments support the claims.

7. The manuscript is about a DNA-based implementation but not a single figure (other than Supplementary Figure 1) include any illustrations with DNA diagrams. The manuscript is missing a high-level introductory figure that can show examples of how features are encoded as DNA, what the secondary structure of similar or dissimilar classes of DNA duplexes would look like, and how those relate to the fluorescence output, which would overall summarize the methods and results of the paper at a glance.

8. There was not enough information provided in the manuscript for me to understand how the encoder and predictor generated similar and dissimilar sequences for each feature and class of images. How were the probabilities of each channel assigned with respect to the input image? If there is a loss function, there should be an expression that the authors can report that tells the reader more about how sequences were decided. Is there an argmax that is being optimized? For any two images, the manuscript currently does not provide me enough information to understand how their DNA sequences would be generated. I believe this is the most critical detail missing from the manuscript and is of utmost importance to address in a revision.

9. There is a lot of prior literature discussed, very verbosely, in the introduction with very little statement of why they are relevant to the paper. It does not appear to me that the authors' architecture relies on strand displacement at all, so the entire paragraph on DNA-based logic and strand displacement seems somewhat unnecessary. Quite frankly, the portion of the paragraph in lines 37-48 could be removed or significantly shortened, while the paragraphs in line 26-36 and 57-73 could also be summarized in some commentary that is no longer than a few sentences in a single paragraph about the parallelization and scale of DNA-based memories and systems. Lines 57-73 could be simply cited as references in the more concise statement on line 62-63 "In recent years, researchers have made significant progress in developing practical and scalable solutions for long-term data storage [citations], such as <and the authors could state here one or two features that represent the most cutting edge features>." Prior works that have direct relation to the authors' work, (e.g. Cherry (2017) and Qian (2011)) that are also classifiers, can be explicitly stated. Get to line 74 and onwards quicker, which is where the discussion relevant to the paper actually seems to begin.

10. In line 87, the authors state there are 500 instances of wet-lab experiments. This is very vague, and I also don't ever see this amount mentioned again. I see '50', as in '50 training samples', was '500' a typo? Otherwise, what is '500' referring to? What exactly was repeated 500 times?

11. A high-level explanation of why the similarity and dissimilarity of sequences matter is not given until line 171 "The largest yield class will be predicted as its class label". This should be stated much earlier. I did not understand what the methods were working towards until I reached this sentence in the manuscript.

12. Line 237&251, the mention of implementing the classifier on a solid surface seems speculative at best, with no references cited, no further details about that implementation, nor evidence for the claim, and it should probably be omitted.

13. I'm not sure what line 256 "massive instances" means or what the intent of the sentence is.

14. The speculation on line 259-262 that the system can scale to a thousand classes seems overzealous. The limitations and scalability of the architecture should be discussed but with more well-supported claims and rationale. The differentiation should depend on the length of DNA sequences, and I would prefer if the authors discussed reasonable limitations or expected scaling factors of the system instead. Such as the lower limit of sequence length needed to differentiate X classes with Y confidence. Intuitively, I would expect that for constant sequence length, as the number of classes increases, the margins between their fluorescent intensity would also become narrower, and thus harder to distinguish.

I suggest the authors also consider citing the following additional literature on DNA-based classifiers.

Lopez, R., Wang, R., & Seelig, G. (2018). A molecular multi-gene classifier for disease diagnostics. *Nature chemistry*, 10(7), 746-754.

Xiong, X., Zhu, T., Zhu, Y., Cao, M., Xiao, J., Li, L., ... & Pei, H. (2022). Molecular convolutional neural networks with DNA regulatory circuits. *Nature Machine Intelligence*, 4(7), 625-635.

Nagipogu, R. T., Fu, D., & Reif, J. H. (2023). A survey on molecular-scale learning systems with relevance to DNA computing. *Nanoscale*, 15(17), 7676-7694.

Yin, F., Zhao, H., Lu, S., Shen, J., Li, M., Mao, X., ... & Fan, C. (2023). DNA-framework-based multidimensional molecular classifiers for cancer diagnosis. *Nature Nanotechnology*, 1-10.

Response to the Review Comments on Manuscript ID: COMMS-23-0213 titled "A nano instance-based learning by non-specific hybridization of DNA sequences"

Dear reviewers,

We sincerely thank you for taking the time to review our manuscript and for providing valuable comments and suggestions.

1. (Reviewer #2) Comment 1:

Author response: We appreciate for this invaluable comment. Motivations of the study are now illustrated in the abstract. (Line 7 to 9)

2. (Reviewer #2) Comment 2:

Author response: Thanks for this valuable feedback and suggestions. We have realized the issue and rewrote the Introduction section. In the new version of the manuscript, this section composed of the following aspects:

- (i) A brief introduction about DNA and DNA data-processing. (Para.1, Line 21 to Line30)
- (ii) A comprehensive introduction about DNA computing, especially, various classifiers implemented using DNA. (Para.2, Line 31 to Line52)
- (iii) Some challenges existing in DNA-based classifiers and the points that the study aims to address. (Para.3, Line53 to Line60)
- (iv) The background about hybridization of DNA sequences, its advantages, applications and disadvantages. Importantly, the main motivation of the study. (Para.4, Line 61 to Line75)
- (v) A general view of the study. (Para.5, Line76 to Line86)

Compared to the last manuscript, we have mainly completed the following improvements as per your recommendations.

- (i) Irrelevant works such as contributions in DNA storage and self-assembly have been streamlined. Notable works in the field of DNA computing, especially in DNA-based classifier have been enhanced to enrich the introduction section.
- (ii) The motivation of the study has been proposed clearly.
- (iii) We have offered the background about hybridization of DNA sequence and the reason why non-specific hybridization is suitable for the study.

These modifications collectively enhance the Introduction section's coherence and its alignment with the goals of the manuscript

3. (Reviewer #2) Comment 3: Results:

Author response: We are grateful for the reviewer's expert guidance. We have carefully considered the comments and have made substantial revisions to the manuscript in Result section, Method section and Supplementary Information section.

- (i) Yield is a measure to quantify the hybridization degree of the DNA sequences. To avoid possible ambiguity, we have revised the definition and calculation of the yield (Line 103 & Line 104), the different ways to estimate yield in different stage (Line 108 to Line 111 & Line 292, in encoder training, by the predictor; in predictor training, by NUPACK).
- (ii) "Hybridization constraint" seems to be an inappropriate expression, we have removed it

from the manuscript. Instead, we used “to map similar/dissimilar images to DNA sequences with high/low yield when one of them is reversely completed” to describe the constraint of the encoder. Descriptions about the encoder training process, the optimization goals and the loss functions of the encoder have been enhanced as well in Method section (Line 299 to Line 337) and Supplementary Note 2.

- (iii) The encoder translates feature vectors into DNA sequences. We have enriched relevant descriptions in both Result section (Line 114 to Line 118), Method section (Line 339 to Line 342 & Line 294), Fig.1b and c.

4. (Reviewer #2) Comment 4:

Author response: The reviewer's feedback has provided us with a valuable external perspective, affirming and enriching our study. We have discussed and reconsidered the limitation of the study, and added the part in Discussion section (Line 270 to Line 277)

5. (Reviewer #2) Comment 5: Final comment:

Author response: We would like to express our sincere gratitude again to the reviewer for the invaluable feedback and insightful comments on our manuscript. Based on these insightful feedbacks, we have diligently revised the manuscript to address their suggestions and concerns.

Reviewer #3 's comments:

1. (Reviewer #3) Comment 1:

Author response: We deeply appreciate reviewer's keen eye for identifying the grammatical and language errors in our manuscript. We take these language issues seriously and have thoroughly reviewed and revised the manuscript accordingly.

- (i) Your valuable feedback and suggestion have brought the grammatical and linguistic errors to our attention. We will carefully review the new version manuscript to avoid similar errors.
- (ii) All “reversal complement” have been replaced by “reverse complement” or “reverse complement sequence” in the new version manuscript. (Line 12, Line98, ...)
- (iii) We have noticed the part of speech issue of the word “hybrid” and “hybridize” and have checked the entire manuscript to avoid similar errors.
- (iv) We have revisited the formatting aspects of our manuscript and have made the necessary adjustments to enhance its visual clarity and professionalism. (Line 35, Line39,...)
- (v) Though the sentence is no longer needed in new version manuscript, we appreciate reviewer's highlights of inappropriate sentences.
- (vi) We have revised the whole manuscript to ensure the usage of abbreviation is both standardized and consistent.
- (vii) The field of synthetic biology is less relevant to the study, we have removed it from the manuscript.
- (viii) We have noticed the grammar issue and revised the manuscript to avoid similar errors.
- (ix) In the revised version of the manuscript, we have improved the exposition concerning the definition and calculation of yield. This enhancement aims to facilitate a more comprehensive understanding of this crucial aspect for the readers.

(x)to (xii) and (xiv) We have noticed the grammar issue and revised the manuscript to avoid similar errors.

(xiii) We have replaced the inappropriate expression.

2. ***(Reviewer #3) Comment 2:***

Author response: We appreciate the reviewer's observation regarding the lack of clarity in the usage of NUPACK. We have elaborated on the technical details in Supplementary Note 3 and Method section (Line 359 to Line 364) to provide a clearer understanding for the readers.

3. ***(Reviewer #3) Comment 3:***

Author response: Thank you for highlighting the concerns related to the unprofessional formatting of our paper. We have taken reviewer's observations seriously and have diligently worked to rectify the formatting inconsistencies (Line385, Line 391, ...).

4. ***(Reviewer #3) Comment 4:***

Author response: Thank you for bringing to our attention the concern regarding unclear descriptions within the paper. We have reexamined and rephrased the manuscript to ensure a more coherent and lucid presentation. In the revised manuscript, we have uniformly referred to the training samples as "instances" and the testing samples as "queries". (Line385, Line 389, ...)

5. ***(Reviewer #3) Comment 5:***

Author response: We are appreciative of your meticulous approach to the review process, which reflects your commitment to maintaining scholarly standards. We have taken immediate action to address the ambiguity you identified and rewrite experiment section to ensure the clarity of our manuscript. (Line 385, Line 389, ...)

6. ***(Reviewer #3) Comment 6:***

Author response: We appreciate the reviewer's feedback regarding the insufficient background knowledge about wet-lab experiments provided in the methodology description. To address the lack of background about SGI, we have included a detailed explanation in Method section to provide readers with a comprehensive understanding. In addition, Method section (Line 344 to Line 357) has been updated to directly state issues about the proposed DNA-based classifier. Including how the DNA-based classifier work, the reason why the SGI is chosen to be a pointer to reflect binding strength, the relationship between fluorescence intensity and binding strength of dsDNA. (Ref. 32&33, Line 373 to Line 378)

7. ***(Reviewer #3) Comment 7:***

Author response: We appreciate the reviewer's comments about the lack of a high-level introductory figure in our manuscript. With the help of the reviewer's invaluable reminder, we have realized the issue and replenished a figure (Fig. 1c) to offer an example about how an image encoding to a DNA sequence. Supplementary Fig.1 has been enriched as well to visualize hybridization between perfectly reverse complement sequences and similar sequences.

8. ***(Reviewer #3) Comment 8:***

Author response: We wholeheartedly agree with this comments that the manuscript lacks of the training detail of the encoder and we wish to express my deep appreciation for their valuable insights. In this revised version, we consider these suggestions to be the primary focus of our modifications, as they play a pivotal role in enhancing the quality and accuracy of the manuscript. In detail:

- (i) Training issues about the feature extractor, the predictor especially the encoder (Method section, Line 299 to Line337) has been extensively revised to offer more information for readers.
- (ii) Loss functions of the predictor and the encoder has now available in Method section to ensure a more professional presentation. Meanwhile, a flowchart illustrating calculation steps of the encoder loss function has been available in Supplementary Note 2 to ensure a more reader-friendly presentation.
- (iii) A comprehensive instance of how an image encode into DNA sequence have been offered in Fig. 1c.

9. (Reviewer #3) Comment 9:

Author response: We appreciate the reviewer's observation regarding the lengthiness of our Introduction section. In response to reviewer's concerns, we have carefully examined our Introduction and removed redundant prior work to improve its overall readability. The central focus of our new version Introduction section lined in the following aspects:

- (i) Background of DNA computing, especially, various classifiers implemented using DNA, which are intricately connected to the focus of our work. (Para.2, Line 31 to 52)
- (ii) Challenges existing in DNA-based classifiers and the points that our study aims to address. (Para.3, Line 53 to Line60)
- (iii) Hybridization of DNA sequences, which is the main tools to accomplish the proposed classifier.

By optimizing the narrative flow, we have succeeded in presenting the Introduction section more succinctly while maintaining its coherence.⁴

10. (Reviewer #3) Comment 10:

Author response: We extend our gratitude to the reviewer for highlighting the typographical errors in our Results section. We have revised typographical errors and related ambiguous sentences in our manuscript to ensure that they now convey our results more clearly.

We conducted a wet-lab experiment comprising 50 instances to verify whether the predicted yield by NUPACK is consistent with the hybridization strength between the query sequence and instances. To mitigate the influence of random errors, the query was systematically repeated three times for each tube containing training samples. (Line 201 to Line 211)

11. (Reviewer #3) Comment 11:

Author response: Thank you for this feedback. We have restructured the narrative of the core proposed method to facilitate readers to comprehend the presented method more comprehensively and at an earlier stage. (Line 101 & Line 102)

12. (Reviewer #3) Comment 12:

Author response: We greatly appreciate the professional guidance provided by the reviewer. We

have added a reference for this idea to make it more convincing. (Ref. 27, Line 254)

13. (Reviewer #3) Comment 13:

Author response: Thank you for this comment. Our intention was to illustrate that the proposed method can be extended to a broader range of instances. We have revised the relevant expressions to ensure clarity and eliminate any ambiguity. (Line 272)

14. (Reviewer #3) Comment 14:

Author response: We extend our sincere gratitude to the reviewer for the expert guidance and insights. We have discussed and reconsidered the scalability and limitation of the study, and added the part in Discussion section.

After careful consideration of your request to discuss reasonable limitations or expected scaling factors of the system, I regret to inform you that due to time limitations, we are currently unable to complete such a discuss as you suggested. Despite our attempts to experiment with shorter sequence lengths, we have not yet approached the lower limit of sequence length. Therefore, we are unable to provide reliable conclusions at this time. We apologize for not being able to fulfill your request. However, we agreed the issue to be a significant direction for future research and discussed in our manuscript (Line 270 to Line 277).

15. (Reviewer #3) Comment 15:

Author response: We deeply appreciate the reviewer's valuable suggestions for additional references, which greatly assisted us during the process of reorganizing the Introduction section. All of these articles have been cited in our manuscript (Line 49, Ref.16; Line 45, Ref.15; Line 53 to Line 60, Ref.18; Line 49 to Line 52, Ref.17). More importantly, we have gained a deeper understanding of DNA-based classifier from them, which will greatly benefit our future research endeavors.

Once again, we are thankful for the opportunity to benefit from your extensive expertise and insights. Your perceptive feedback has undoubtedly elevated the quality and comprehensiveness of our manuscript.

Warm regards,
The authors.

REVIEWERS' COMMENTS:

Reviewer #2 (Remarks to the Author):

Thank you for considering and addressing all of my comments. I am very happy to accept this work.

Reviewer #3 (Remarks to the Author):

The reviewer appreciates the clear effort from the authors to address the previous comments. The edits to the manuscript are excellently done. All comments have been fully addressed. The reviewer supports the publication of the manuscript.

There are several additional minor revisions the authors may consider, should they agree with them, that do not impact my recommendation for publication. I will defer to the authors and the editorial team to decide which of these may be considered necessary or not.

1. Suggestions for some sentences that could be improved:

1.i) Line 71, "The main advantages of this learning 'is' that 'it' is..."

1.ii) Line 75, I suggest "at molecular 'scale'" or "'by' molecular 'computation'" instead of "at molecular scenario".

1.iii) Line 76, "...demonstrate 'the construction of' an instance-based..." instead of "...demonstrate to construct an instance-based..."

1.iv) Line 115, "...maps a 50-dimensional (50-D) feature vector into a." It seems like this is missing a word at the end. "...into a 4 x 59 tensor."?

1.v) Line 150, rather than "four pairs of images" it seems Fig 2c actually shows "four pair of images for each interval".

2. Some improvements to Fig 1a would make it a much better figure:

2.i) Could the parts of the workflow that are actually DNA or DNA sequences be drawn as DNA (just simple line arts would be acceptable, i.e. a line for the backbone and extending lines for each nucleotide) instead of simple rectangles?

2.ii) Also to consider: write example data in each rectangular block. Not fully, but just a few values as an example of the data type in that block. i.e. You do not have to write out a 59 character string for a 59-nt DNA sequence, just a few letters "ATGCG...". Pixel data blocks can have numbers between 1-256 for RGB pixel values.

2.iii) Grid lines within each block so that the dimensions are more apparent. The scales of the blocks feel random and not indicative of the data they represent (16x8x8 and 16x4x4 blocks sizes are very different, while an 8x55 block is the same width as a 1x53 block). There is obviously not enough space to make a 1x1 data block look overly different based on size alone to a 1x53 data block, but adding grid lines within each of the blocks will help to differentiate their scale.

2.iv) 3D data should be drawn as such (cubes).

Dear Reviewer #3,

We would like to express our sincere gratitude to you for your meticulous review of our manuscript, titled " *Nano scale instance-based learning using non-specific hybridization of DNA sequences* " (ID: COMMS-23-0213), and for providing invaluable feedback. Your consideration and thoughtful comments have significantly contributed to improving the quality of our work. We have carefully incorporated your suggestions, and the specific revisions listed as followed, revised manuscript can be found in the attached document.

We have addressed and implemented all of your Suggestion #1, and we have conducted a thorough review to avoid similar errors elsewhere.

- i) Line 73-75;
- ii) Line 77;
- iii) Line 78&79;
- iv) Line 117;
- v) Line 145-147.

Your second suggestion has provided us with valuable inspiration. We have made thorough revisions to the content of Fig.1 in accordance with your recommendations:

1. DNA Sequence Issue: Considering that Fig.1a represents the main architecture of our method, and DNA sequences are represented by 4×59 tensors, we have added tensors to Fig.1a to make the output of the encoder and the input of the predictor clearer. After discussions, we believe that inserting DNA molecule representations in Fig.1c would be more appropriate. Additionally, we have adjusted the color scheme in Fig.1c to ensure a more consistent style throughout the entire figure.
2. Block Size Issue: We acknowledge that the block sizes in our last version were disproportionate to the tensor sizes, potentially leading to misunderstandings. To facilitate visualization and layout considerations, we have retained the exact same block sizes in the revised manuscript. We have also clearly annotated the output sizes and parameters of each block ensuring that the entire figure is rich in information and maintains a consistent style.

We hope that these modifications align with your expectations. Once again, we sincerely appreciate your valuable insights and your time.

Warm regards,
The authors.